# Impact of warming shelf waters on ice mélange and terminus retreat at a large SE Greenland glacier

Suzanne L. Bevan[1], Adrian J. Luckman[1], Douglas I. Benn[2], Tom Cowton[2], and Joe Todd[2]

[1]Swansea University, Singleton Park, Swansea SA2 8PP
[2]University of St Andrews, College Gate, St Andrews KY16 9AJ

**Correspondence:** S. L. Bevan (s.l.bevan@swansea.ac.uk)

**Abstract.** By the end of 2018 Kangerdluqssuaq Glacier in southeast Greenland had retreated further inland than at any time in the past 80 years and its terminus was approaching a region of retrograde bedslope from where further rapid retreat would have been inevitable. Here we show that the retreat occurred because the glacier failed to advance during the winters of 2016/17 and 2017/18 owing to a weakened proglacial mélange. This mixture of sea ice and icebergs is normally rigid enough to inhibit calving in winter but for two consecutive years it repeatedly collapsed allowing Kangerdlugssuaq Glacier to continue to calve all year round. The mélange break-ups followed the establishment of anomalously warm surface water on the continental shelf during 2016 which likely penetrated the fjord. As calving continued uninterrupted from summer 2016 to the end of 2018 the glacier accelerated by 35% and thinned by 35 m. These observations demonstrate the importance of near-surface ocean temperatures on tidewater glacier stability, and show that it is not only deep-ocean warming that can lead to glacier retreat. During winter 2019 a persistent mélange reformed and the glacier readvanced by 3.5 km.

## 1 Introduction

Since the early 1990s the Greenland Ice Sheet (GrIS) has been a major contributor to sea-level rise, losing a total of 2700±930 Gt of ice between 1992 and 2011 (Shepherd et al., 2012). About 40% of the 0.47±0.23 mm/year mean 1991–2015 sea-level rise originating from Greenland was caused by increases in the rate at which glaciers calve ice into the oceans, and the remainder by increases in surface melt and runoff (van den Broeke et al., 2016). Kangerdlugssuaq Glacier (KG) is a large tidewater-terminating glacier in southeast Greenland (Fig. 1) which delivers around 24 Gt/year of ice to the ocean, equivalent to about 5% of GrIS total discharge (Enderlin et al., 2014). In 2005 the calving front of KG rapidly retreated by over 6 km, its surface flow speeds doubled (Howat et al., 2005; Luckman et al., 2006), and between 2003 and 2007 the glacier thinned by over 100 m (Khan et al., 2014). The rapid retreat of KG was accompanied by a similar pattern of change in many other southeast Greenland outlet glaciers and accounted for ∼16% of the total 2000–2005 net mass loss of the GrIS (Rignot and Kanagaratnam, 2006). KG thus typifies the response of Greenland outlet glaciers to climate forcing whilst being an individually important source of sea level rise. After 2006, KG slowed down, although speeds remained at least 20% greater than pre-retreat values, and the ice front maintained a steady mean annual position, with seasonal advances and retreats of up to 6 km (Kehrl et al., 2017).

The synchronous retreat of southeast Greenland glaciers in the early 2000s suggested that atmospheric and/or ocean warming were responsible for initiating the rapid retreat, thinning and subsequent dynamic response (Howat et al., 2008; Hanna et al., 2009; Murray et al., 2010; Howat and Eddy, 2011; Christoffersen et al., 2011; Inall et al., 2014). Nevertheless, local differences in glacier and fjord geometry, and connection to the ocean normally determine individual responses to changing environmental

conditions (Moon et al., 2012; Enderlin et al., 2013; Millan et al., 2018). When KG retreated in 2005 it did so into deeper water (Khan et al., 2014) — such a reverse or retrograde bed slope can set up a positive feedback between frontal retreat, ice discharge and dynamic thinning (Schoof, 2007). KG's summer maximum advance position continued to retreat up-fjord gradually until mid 2011 when its grounding line once again reached shallower water and the glacier began to slow down (Kehrl et al., 2017). At that stage thinning had also slowed and the final few km of the glacier was afloat (Khan et al., 2014; Kehrl et al., 2017).

KG calves into the head of a 75 km long, 5–10 km wide, fjord (Murray et al., 2010; Sutherland et al., 2014) (KF); the fjord has a wide mouth and is connected to the shelf break by a deep, straight, 300 km long trough (Dowdeswell et al., 2010; Inall et al., 2014). Much of the passage from ocean to glacier is 600–900 m deep with sills shallowing to 400–550 m at the fjord mouth and within the shelf trough (Fig. 1). The increase in mass loss from the southeast GrIS to the ocean that began in the mid 1990s coincided with a warming of the North Atlantic Ocean (Straneo and Heimbach, 2013) and the relatively unimpeded

connection of KG with the ocean was considered to have allowed increasing ocean temperatures to trigger retreat in 2005 (Christoffersen et al., 2011; Inall et al., 2014; Jackson et al., 2014; Millan et al., 2018).

More recently, following two winters when KG unusually failed to advance, its calving front by the end of 2018 was further retreated than at any point during the observational record (i.e. since 1932, Brough et al., 2019). We present evidence that the recent retreat was triggered by a weakening of ice mélange in the fjord, a mechanism only previously shown by association

(Moon et al., 2015). We propose that the mélange weakening is explained by exceptionally warm surface waters originating from outside the fjord. The observed contemporaneous interannual thinning superimposed on the seasonal cycle of surface elevation change will leave the glacier vulnerable to basal melt and further rapid retreat.

## 2  Methodology and Data

We use time series of satellite imagery, feature tracking of SAR and optical data, interferometric digital elevation models

(DEMs), and ocean and air temperatures to place the recent retreat in the context of the multidecadal satellite record and to explain the cause of this retreat.

### 2.1  Frontal positions

We manually digitised glacier fronts on a variety of optical and synthetic aperture radar (SAR) satellite images. We located the intersection points of the digitised fronts with a series of parallel linear flowlines at 160 m spacing, and the frontal change was

calculated by taking the mean of the changes in these intersection points (Luckman et al., 2015). From 1985–2012 the images include Landsat-5 (TM Band 4) and Landsat-7 (ETM+ Band 8), European Remote Sensing satellites (ERS-1 and ERS-2), and Envisat Advanced Synthetic Aperture Radar (ASAR) Image (IM) and Wideswath mode (WSM) data (Bevan et al., 2012). From

2011–2018 we use TerraSAR-X SAR data, and additionally, from 2015–2019, Sentinel 1A and 1B Ground Range Detected High-resolution Interferometric Wideswath (GRDH, IW) images. Image spatial resolution ranges from 30 m for Landsat-5 to 8 m for the multi-looked TerraSAR-X data. All images were reprojected to the Polar Stereographic co-ordinate system before the fronts were digitised. Appendix Fig. A1 shows the observations derived from different satellite missions.

## 2.2 Surface velocities

We used feature tracking to derive surface velocities — see Bevan et al. (2012) for details of the early (1985–2012) part of the time series. After 2011 velocities are based on TerraSAR-X SAR and Sentinel 1A and 1B single-look complex (SLC) data using Gamma Remote Sensing software. Pairs of TerraSAR-X SLCs with 11 day time separation were tracked using a window spacing of 40 m and the results converted to ground range and geocoded using coincident interferometric DEMs (Section 2.5). Sentinel-1 pairs were tracked with pair delays of either 6 or 12 days, with a window spacing of 100 m and geocoded using the 90 m Greenland Icesheet Mapping Project (GIMP) DEM (Howat et al., 2014). As for frontal positions, Appendix Fig. A1 shows the velocity observations derived from different satellite missions.

## 2.3 Ice mélange

Similarly to Kehrl et al. (2017), we produce a metric for the existence of rigid ice mélange based on the ability of feature tracking to capture velocities in the region immediately in front of the glacier terminus. If a visual inspection of mapped velocity magnitude shows a realistic, coherent and uniform velocity field extending across the width of the fjord and for at least 5 km downstream of the glacier front we determine there to be rigid mélange present. In summer, when the glacier is calving, it is very rare for feature tracking to capture realistic surface velocities in the fjord, indicating that rigid mélange does not persist between consecutive images. This method is an improvement on Kehrl et al. (2017) in that it assimilates information from the whole fjord rather than sampling the coherence at a single point. Note that the feature-tracking algorithm looks for a two-dimensional translation of features over the time period between successive images (11 days for TerraSAR-X, 6 or 12 days for Sentinel 1) meaning that the mélange must last at least this long to be tracked.

## 2.4 Ocean and surface air temperatures

We extracted ocean potential temperatures for 1991–2017 from Arctic Ocean Physics Reanalysis monthly mean data supplied by the Copernicus Marine Environment Monitoring Service (CMEMS). Temperatures for 2018 and into 2019 were then based on monthly means of the Arctic Ocean Analysis and Forecast Product (also supplied by CMEMS) which are available daily. Both the reanalysis and the analysis/forecast products are based on a 3D physical ocean and sea-ice model that assimilates remotely sensed and in-situ data, and have a spatial resolution of 12.5 km, and 12 depth levels distributed unevenly between 5 and 3000 m. The data do not extend into KF but we calculated the mean, standard deviation and monthly anomalies of the 5 m and 200 m potential temperatures over the Kangerdlugssuaq trough area (Fig. 1) for the full time series. The 5 m data

were chosen to sample Polar Surface Water (PSW) and the 200 m data to sample the upper layers of Atlantic Water (AW) (Sutherland et al., 2014). Within the fjord we rely on published Conductivity Temperature Depth (CTD), for context.

We downloaded 2 m air temperatures from the Danish Meteorological Institute weather reports for Aputiteeq (station number 04351) at the entrance to KF (see Fig. 1). Anomalies in monthly mean 1200 UTC temperatures were calculated relative to the period 1987–2018; 1987 being the first year for observations at 04351.

## 2.5 Surface elevation

We derived a time series of 150 DEMs from June 2011 to July 2018 using experimental SAR data from the TanDEM-X satellite system which comprises the TerraSAR-X and TerraSAR-X add on for Digital Elevation Measurement (TanDEM-X) satellites. We used Gamma Remote Sensing software to interfere, unwrap and geocode the bistatic stripmap mode Co-registered Single look Slant range Complex images (CoSSCs). The CoSSCs have a spatial resolution of ∼2 m and the DEMs were smoothed to a horizontal resolution of 8 m. We used the provided orbital vector data and the 30 m GIMP DEM (Howat et al., 2014) to initially geolocate and phase scale the images; geolocation was iteratively improved using the interferometrically generated DEMs. The interferograms were unwrapped from a bare-rock location on the south side of the glacier (33.0365°W,68.5939°N), and the DEMs vertically tied to this point using the GIMP DEM height (730 m). Only CoSSCs with satellite separations perpendicular to the look direction of less than 500 m were used because longer baselines prevented satisfactory unwrapping. This restriction meant that there were fewer DEMs created for 2015. Elevations are given relative to the WGS84 reference ellipsoid. Orbit uncertainties (Krieger et al., 2013) mean that we cannot expect relative elevation accuracies better than 2 m. The standard deviation of DEM heights at a point on the opposite side of the fjord to the unwrapping start point was 2.3 m, indicating that unwrapping errors were minimised even across the glacier. The accuracy of absolute height values depends on the accuracy of, and geolocation with respect to the GIMP DEM. The GIMP DEM in turn is quoted as having a vertical precision of between ±1.0 m over most ice areas and ±30 m over areas of high relief (Howat et al., 2014). We therefore estimate absolute errors of the order of ±10 m which is the root mean squared validation error of the GIMP DEM with respect to ICESat.

For the bed we used IceBridge BedMachine Greenland, Version 3 data (Morlighem et al., 2017) together with the hydrostatic equilibrium assumption to determine where the glacier surface was above flotation height. We used an ice density of 917 kg/m$^3$, and a sea-water density of 1023 kg/m$^3$. At the location of KG, WGS84 datum is 55 m below the Geoid.

## 3 Results

### 3.1 Ice front position

Our long and detailed record of ice-front position shows that prior to the 2005 retreat and from 2005 to the end of 2016, KG maintained a relatively stable mean-annual frontal position (Fig. 2). The clear seasonal variation in ice-front position is characterised by an advance of between 2 and 6 km from January until July or August (which we will refer to simply as winter), with almost no calving events. During the second half of the year (summer) the front normally retreats in a steady manner.

By contrast, in 2017 and 2018, the glacier continued to calve throughout both summer and winter, only advancing a fraction of the normal distance between the beginning of January and the end of July in 2017, and not at all in 2018 (Fig. 3a). This lack of sustained winter advance has only occurred twice before in the observed record: 1996, and 2005 which marked the previous episode of retreat and thinning (Fig. 2, Luckman et al., 2006). May 2011 also featured an unusually early period of calving. By the start of summer 2018 the ice front had retreated by 8 km relative to the start of summer 2016 and the anomalous winter calving continued into the summer months. In 2019, after two calving events in January, the ice front continued to advance as normal for this time of year. By the end of May 2019 the ice front was 3.3 km upstream of its position at the start of summer 2016.

## 3.2 Surface velocities

The increasingly high temporal resolution record of surface velocities shows that the peak in 2005 remains a record for Kangerdlugssuaq but that velocities coincident with the more recent retreat were as high as they have ever been since 2005. At the start of 2017, when the normal winter advance faltered, velocities increased and the acceleration was sustained through to summer 2018, by which time KG was flowing 35% faster than two years earlier (Fig. 3a). In 2019 as KG readvanced surface velocities slowed. Both secular and seasonal retreats (such as 1996), and individual calving events result in glacier acceleration.

## 3.3 Integrity of proglacial mélange

We assessed 254 TerraSAR-X and Sentinel 1 velocity maps between 13/02/2012 and 30/05/2019 for evidence of rigid proglacial mélange (see, e.g. Fig. 4). It is evident (Fig. 3a) that in 2012, 2013, 2016 and 2019 glacier advance coincided with sustained periods of rigid mélange. In winters 2017 and 2018, when the glacier continued to calve, a rigid mélange was not maintained for more than one month in either year.

### 3.3.1 Air and ocean temperatures

Air temperatures were anomalously warm throughout 2016 relative to the 1987–2018 mean, most notably between September 2016 and February 2017 (Fig. 3b). Similarly, exceptionally mild air temperatures were recorded between December and April of the following winter. Earlier in the record (Fig. 2d) temperatures were significantly above the 1987–2018 mean in December 2002 ($+6.7°$C), March 2005 ($+5.5°$C), and in January 2014 ($+7.3°$C).

Surface waters on the southeast Greenland continental shelf were also exceptionally warm in 2016: by July and August 2016 potential temperatures at a depth of 5 m were up to $4°$C warmer than the 1992–2018 mean (Appendix, Fig. A2) and the anomalies persisted over winter and into the first half of 2017 (Figs. 2b and 3b). Of note also is the anomalously warm water at 5 m and 200 m in the winter of 2002/2003. Since 2012 winter temperatures at 200 m have been steadily increasing (Fig. 2c).

CTD data (Fig. 5) provide in-fjord temperature data for autumns of 1991, 1993, 2004, 2009, 2010 and 2017 from the surface down to depths of 400–700 m. The profiles show relatively warm water at depth in September 2010 which is reflected in the

re-analysis data at 200 m (Fig. 2c). However, whilst the October 2017 profile shows warm temperatures in the upper 100 m, the waters at depth are not as notably warm as the re-analysis data show out on the shelf.

## 3.4 Surface elevation and evolution of a floating tongue

Associated with the seasonal velocity pattern is a seasonal dynamic change in KG surface elevations — the glacier thins as it retreats and accelerates, and thickens as it advances and slows (Figs. 3a and c). During the recent period of acceleration (June 2016 to May 2018) a cross-glacier mean thinning of 35 m is superimposed on the seasonal thinning. Surface elevations relative to sea-level and an assumption of hydrostatic equilibrium indicate that in summer 2016 the final 5 km of the glacier was floating (Fig. 7); the velocity profile for 04/06/2014 (when the glacier thickness and front location were similar to summer 2016) confirms this. The reduction in down-flow acceleration between kilometers 11 and 4 is consistent with transition to a floating tongue and loss of basal drag. By May 2018 (the last available DEM) most of this floating tongue had been lost and the ice front was left only 1 km seaward of the start of a section of reverse bed slope (Figs. 6 and 7).

## 4 Discussion

Based on the success or otherwise of the feature-tracking method to capture velocities over the region immediately in front of KG's terminus we have confirmed that any winter advance coincides with the formation of a rigid mélange (Fig. 3a). Although interstitial sea ice likely contributes little to mélange strength, it prevents iceberg dispersal and thus encourages the transfer of back stress from the fjord sides to the glacier front via compressional stress bridges between adjacent icebergs (Burton et al., 2018). Back stress from mélange is clearly insufficient to resist advance of the glacier front, but it inhibits calving by reducing crevasse propagation and preventing the detachment of icebergs, even where the ice is heavily fractured (Cassotto et al., 2015; Fried et al., 2018). It has been demonstrated theoretically that the amount of back stress necessary to prevent calving is of the order of $10^7$ N/m and that this can be readily supplied by ice mélange (Amundson et al., 2010; Krug et al., 2015).

At KG the mélange inhibits detachment of icebergs from the glacier front until sea ice melts and the mélange disperses with the onset of summer — resulting in the seasonal advance/retreat cycle. The magnitude of this cycle can be explained by modelling studies that have shown that the inhibiting effect of ice mélange on calving is capable of generating much larger (km scale) seasonal advance and retreat cycles than the annual variation in submarine frontal melt (Todd and Christoffersen, 2014; Krug et al., 2015; Todd et al., 2018) .

In contrast to previous years, in early 2017 and 2018, formation of a rigid mélange was repeatedly interrupted by episodes of break-up and dispersal. Close examination of a series of synthetic aperture radar (SAR) images from Sentinel 1 reveals that each mélange breakup episode commenced at the down-fjord edge and propagated towards the glacier front, culminating in large calving events (Fig. 8, and Video 1). In consequence, the normal sustained advance during winter was punctuated by several periods of calving. This behaviour is similar to that usually experienced during the last 6 months of each year, when rigid mélange repeatedly forms and breaks up. The failure of KG to advance in the winters of 2017 and 2018 thus reflects weakly bonded mélange and indicates that conditions in the fjord were not conducive to the formation of mélange at this time.

The overarching cause of retreat may therefore lie in a warming of air temperatures, ocean temperatures, or a combination of these effects.

Mild air temperatures, particularly during the winter months, could contribute to the weakening of ice mélange by delaying or limiting sea ice formation. The anomalously warm winter air temperatures of 2016 and 2017 suggest that atmospheric warming played a role in driving recent retreat at KG. Similarly, air temperatures in early 1996 and 2005, both years when there was a lack of winter advance, were very warm. However, the warm air temperatures in 2014 had no apparent impact on the winter advance of KG that year. It may therefore be that warm winter air temperatures alone are not necessarily sufficient to inhibit mélange formation without concurrent favourable ocean or atmospheric circulation patterns.

The warming of shelf waters in July and August 2016 in part reflects the high regional air temperatures during this time, but was also driven by an increased advection of water from the Atlantic Ocean (Timmermans , 2016). If these exceptionally warm surface waters were able to propagate into the fjord, then they may have contributed significantly towards the weakened ice mélange at KG during winter 2016/17. Although there is no record of water properties within KF in 2016, observations from previous years (Inall et al., 2014; Sutherland et al., 2014), and numerical modelling experiments (Cowton et al., 2016), indicate that seasonally warmed shelf surface waters (termed Polar Surface Water warm, PSWw) can readily penetrate far into KF. These studies reveal the prevalence of a complex, multi-layered circulation in KF during the summer melt season. Turbulent upwelling, forced by the input of meltwater at the glacier termini (and in particular KG), creates an outflowing tongue of relatively cool glacially modified waters (GMW) at a depth of ∼100 m, with a compensatory up-fjord flow of warm, salty AW below this. Above the GMW, circulation is characterised by a secondary cell in which PSWw is drawn towards the fjord head, capped by a thin (∼10 m) layer of outflowing cool, fresh water. Numerical modelling (Cowton et al., 2016), and analysis of these waters in temperature–salinity space (Inall et al., 2014), indicate that this surface outflow is forced by the input of freshwater from surface runoff and shallow tidewater glaciers. The upper cell therefore resembles a classic estuarine circulation, with the up-fjord flow of PSWw driven by turbulent mixing at the interface between these overlying layers.

Observations show that PSWw may constitute the warmest water mass in KF during the summer months (Inall et al., 2014). Although it experiences cooling through mixing and iceberg melt during up-fjord transit, available hydrographic data from the inner fjord demonstrate it remains relatively warm ($\sim 0.5 - 1.5°C$) even at this distance from the shelf (Fig. 5). Thus, while considerable attention has been given to AW as a driver of submarine melting (Straneo and Heimbach, 2013), PSWw represents an important yet comparatively overlooked component of the fjord heat budget. Inall et al. (2014) calculated that PSWw accounts for 25% of ice melt within the fjord system reflecting its warmth and proximity to floating ice within the fjord. Indeed, its location near the fjord surface means that it could inhibit mélange formation. We therefore propose that exceptionally warm PSWw played an important role in the weakening of the ice mélange, and thus onset of retreat, at KG in 2016/17.

In addition to the anomalously warm shelf surface waters, it is possible that a warming of subsurface AW may have contributed to the retreat of KG. These subsurface waters, unhindered by the relatively deep sills, are advected into KF by both the buoyancy-driven circulation described above and by coastally trapped waves associated with winter storms (Jackson et al., 2014; Cowton et al., 2016; Fraser and Inall, 2018). The retreat of KG and other Greenlandic tidewater glaciers has been asso-

ciated with a warming of coastal AW (Straneo and Heimbach, 2013; Cowton et al., 2018), although the difficulty of observing

frontal processes at tidewater glaciers means that evidence of causation remains elusive.

While warming AW may have helped precondition KG for retreat, there are several reasons why we believe it is unlikely to have been the principal cause of the retreat commencing in 2016. Firstly, the onset of retreat in winter, and observations of reduced mélange rigidity immediately prior to calving, indicate that the retreat was triggered by a weakening of the mélange, most likely due to reduced sea ice formation. While the subsurface input of meltwater at KG drives vigorous upwelling of AW,

observations and modelling indicate that this reaches neutral buoyancy and flows out of the fjord at depths greater than $\sim$50m (Inall et al., 2014; Cowton et al., 2016). At these depths it may contribute to melting of larger icebergs in the mélange, but is less well placed to impact on sea ice formation at the fjord surface, and hence mélange rigidity. Secondly, while reanalysis data indicate that AW in Kangerdlugssuaq trough has warmed in recent years, this warming has been relatively gradual, with no obvious trigger for retreat in 2016/17. This contrasts with surface waters, which experienced exceptionally high temperatures in

2016. Thirdly, a hydrographic profile obtained from KF during autumn 2017 does not show particularly warm AW in the fjord at this time (Fig. 5). This implies that the apparent warming of trough waters may to some extent be an artefact of the reanalysis data, or may not be effectively translated into an actual warming of AW in KF, especially in the autumn when buoyancy-driven circulation is weaker.

We therefore propose that the exceptionally warm surface waters on the east Greenland shelf during 2016 and early 2017,

and consequently the presence of anomalously warm PSWw within KF, played a critical role in triggering the recent retreat of KG. Combined with mild air temperatures, we suggest these warm, near-surface waters will have acted to delay and weaken winter sea ice formation, reducing mélange rigidity and thus allowing increased winter calving of KG. The possibility that anomalously warm PSWw could be responsible for episodes of retreat at KG has been previously hypothesised; Christoffersen et al. (2011) noted that CTD data from inner KF showed PSWw at $> 2°C$ shortly prior to the major retreat of 2005, compared

with $< -1°C$ during a previous survey during the stable year of 1993, and observed a corresponding break up of mélange in satellite imagery. Christoffersen et al. (2012) presented evidence that warm air temperatures in early 2005 (see also Fig. 2d) together with strong katabatic winds contributed to the reduction in the mélange in front of KG. According to Christoffersen et. al. the katabatic winds act to drive the sea ice away from the glacier front and were weaker in 2002/03 when KG did not retreat in spite of warm air and shelf ocean temperatures (Fig. 2). The lack of winter advance in 1996 was accompanied by

above average shelf ocean temperatures as well as above average air temperatures.

Warm air temperatures may then help to explain the continued retreat during 2017/18 even when shelf surface waters were not observed to be anomalously warm although the CTD data for October 2017 show warm water in the fjord in the upper 100 m. Air temperatures in early 2018 were almost $6°$ above average and the mélange broke up on a number of occasions. It was not until 2019 that sea surface temperatures fell below the mean and a continuous spell of rigid mélange allowed the

terminus to readvance even though 200 m temperatures remained high on the shelf (Figs. 2c and 3b).

## 4.1 Implications of further retreat

A decrease in surface elevations of outlet glaciers in the early 1990s was one of the first indications that the GrIS was losing mass via increased surface melt and dynamic thinning (Krabill et al., 2000; Krabill, 2004). Dynamic thinning is the result of acceleration when retreat and melt-driven thinning reduce resistive stresses at the glacier front. Seasonal cycles of dynamic thinning have recently been observed on Helheim, another large tidewater glacier 300 km to the south of KG (Bevan et al., 2015), where they are associated with fluctuations in ice-front position (Kehrl et al., 2017). We also see dynamic thinning on KG (Fig. 3c).

The two-year retreat of KG to a point only a kilometre or so seaward of the reverse bedrock slope, and further up fjord than at any point in the observation record, placed the glacier in a precarious position. Continued retreat, acceleration and dynamic thinning would have resulted in the ice front refloating when it would then have been susceptible to basal melt and further thinning. However, during the first 5 months of 2019 the glacier advanced as normal during the winter and the mélange remained intact. Air temperatures are not yet available but shelf near-surface temperatures remained below average between December 2018 and April 2019 (Fig. 3b). Thus the behaviour of KG remains consistent with the idea of mélange back stress being the dominant control on frontal evolution.

## 5 Conclusions

We conclude that the retreat of KG in 2017 was caused by weakened winter ice mélange that allowed sustained calving when the glacier would normally be advancing. The mélange is likely to have been weakened initially by anomalously warm water on the shelf in the latter half of 2016 which likely penetrated the fjord, and by anomalously warm air temperatures. Continued retreat in 2018 was facilitated by warm air temperatures hindering the formation of sea ice to bind a rigid mélange. Any additional retreat of KG would have taken the terminus into a region of retrograde bed slope, which could have resulted in further retreat and thinning via dynamics and basal melt. Cooler water on the shelf and presumably in the fjord during winter 2019 allowed a persistent mélange to form and the glacier front readvanced by 3 km.

Our research emphasises the importance of accounting for the delivery of heat into the fjords of Greenland by surface as well as deep water. Heat delivered by surface water can weaken the stabilizing influence of ice mélange proximal to ocean-terminating glaciers, disrupting the seasonal calving pattern and triggering terminus retreat.

*Data availability.* The NERC Polar Data Centre hosts the timeseries data on frontal positions (https://doi.org/10.5285/b317f707-2ef6-449c-acc3-6bb087efecb1), surface velocities (https://doi.org/10.5285/c26e3873-e33e-45be-b76b-87f3b8827101) and surface elevations (https://doi.org/10.5285/3bbacca6-d2cd-46be-b824-b828572ca486), and Video 1 (https://doi.org/10.5285/61100705-dfbc-489d-b729-1268ec743bbf).

*Author contributions.* The study was led and the manuscript drafted by Suzanne Bevan, the project was led by Doug Benn and Adrian Luckman. Expert advice, data and contributions to the text and figures were supplied by Adrian Luckman (SAR imagery and Sentinel velocities), Douglas Benn and Joe Todd (interpretation of glacier calving and interaction with mélange), Tom Cowton (fjord processes). TanDEM-X DEMs and TerraSAR-X velocities were processed by Suzanne Bevan.

*Competing interests.* The authors declare that they have no competing interests.

*Acknowledgements.* The research under project CALISMO (Calving Laws for Ice Sheet Models) was funded by the Natural Environment Research Council (NERC) grants NE/P011365/1. TanDEM-X data used for generating the DEMs and for ice-front positions and glacier velocities were supplied by DLR, as part of NERC project NE/I0071481/1. Satellite imagery for front positions were supplied by European Space Agency (ERS-1 and -2, Envisat, Sentinel), the U.S. Geological Survey (Landsat). BedMachine data were obtained from NASA National Snow and Ice Data Center Distributed Active Archive Center. doi: https://doi.org/10.5067/2CIX82HUV88Y. [16/01/2018]. DMI weather observations were downloaded from www.dmi.dk/publikationer. The CMEMS ocean reanalysis data was downloaded from ftp://mftp.cmems.met.no/Core/ARCTIC_REANALYSIS_PHYS_002_003/dataset-ran-arc-myoceanv2-be, and the ocean analysis and forecast data from ftp://mftp.cmems.met.no/Core/ARCTIC_ANALYSIS_FORECAST_PHYS_002_001_a/dataset-topaz4-arc-myoceanv2-be.

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

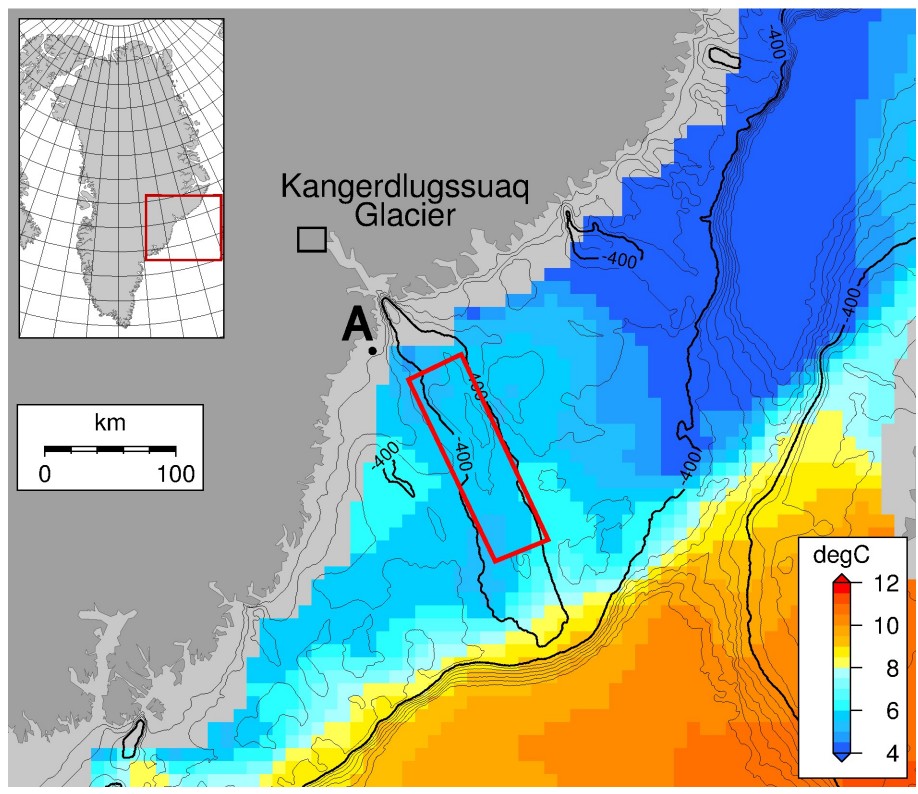

**Figure 1.** South-east Greenland with reanalysis 5 m ocean temperatures for July 2016. The bathymetric contours are every 100 metres and are based on the International Bathymetric Chart of the Arctic Ocean Version 3 (Jakobsson et al., 2012). Small black box shows the Kangerdlugssuaq Glacier area covered by Fig. 6. Red box shows the area over which ocean temperatures were averaged for Figs 2 and 3. Aputiteeq weather station is marked by the letter A.

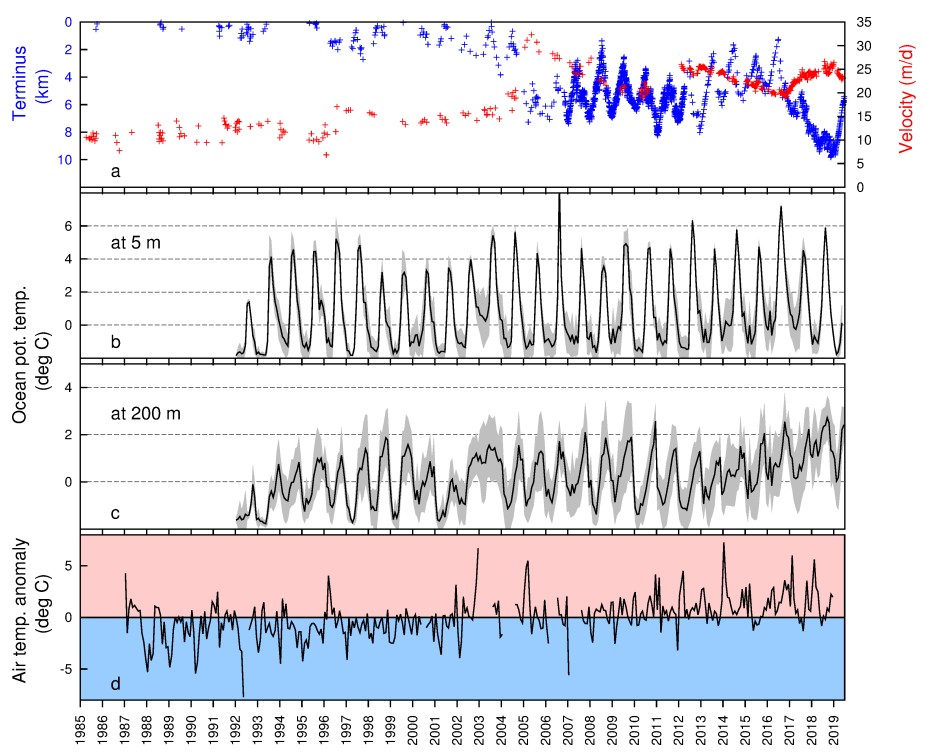

**Figure 2.** Time series of a) Kangerdlugssuaq Glacier front position and surface velocity. The y-axis distance scale for the front position matches the profile drawn in Fig. 6. Front positions and velocities are based on a variety of satellite images and the velocities were measured at the star marked in Fig. 4. Appendix Fig. A1 indicates which instruments contributed which observations. b) Kangerdlugssuaq trough monthly ocean potential temperature at 5 m averaged over the red box shown in Fig. 1. The grey shade shows ± one standard deviation around the mean. c) As for b) but at 200 m. d) Anomalies in monthly mean 12z air temperature at Aputiteeq.

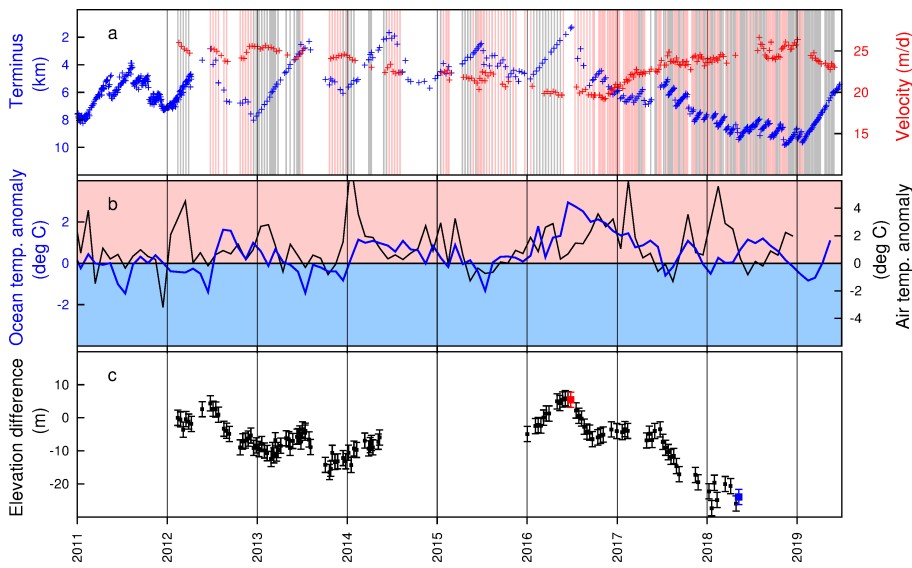

**Figure 3.** Time series plots. a) Front position and surface velocity (note the change in front position and velocity scales compared with Fig. 2a). The vertical bars mark dates when the feature-tracking metric indicated the presence (grey) or absence (pink) of a rigid mélange. b) 2 m air temperature anomaly for Aputiteeq, and Kangerdlugssuaq trough ocean potential temperature anomaly at 5 m depth averaged over the red box shown in Fig. 1 (blue). c) Cross-glacier mean surface elevation difference taken from the TanDEM-X DEMs, the error bars represent the relative accuracy of $\pm 2.3$ m. The elevations are an average across the transect marked in Fig. 6, and the differences are relative to the first DEM. The red and blue points mark the DEMs used for the surface elevation profiles plotted in Fig. 7.

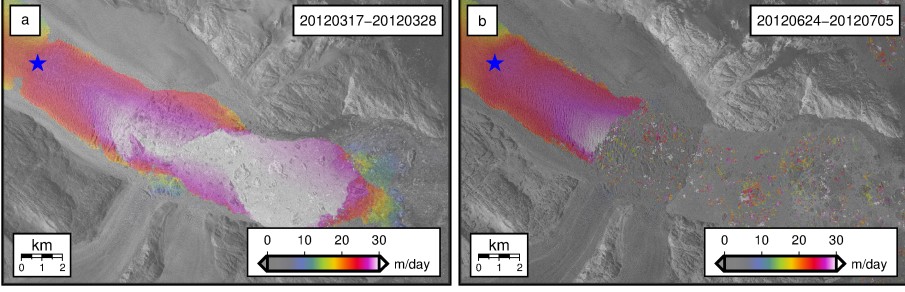

**Figure 4.** Feature-tracked surface velocities based on TerraSAR-X data to demonstrate discrimination between a) rigid and b) non-rigid mélange. The blue star marks the extraction point for the velocities presented in Figs. 2 and 3.

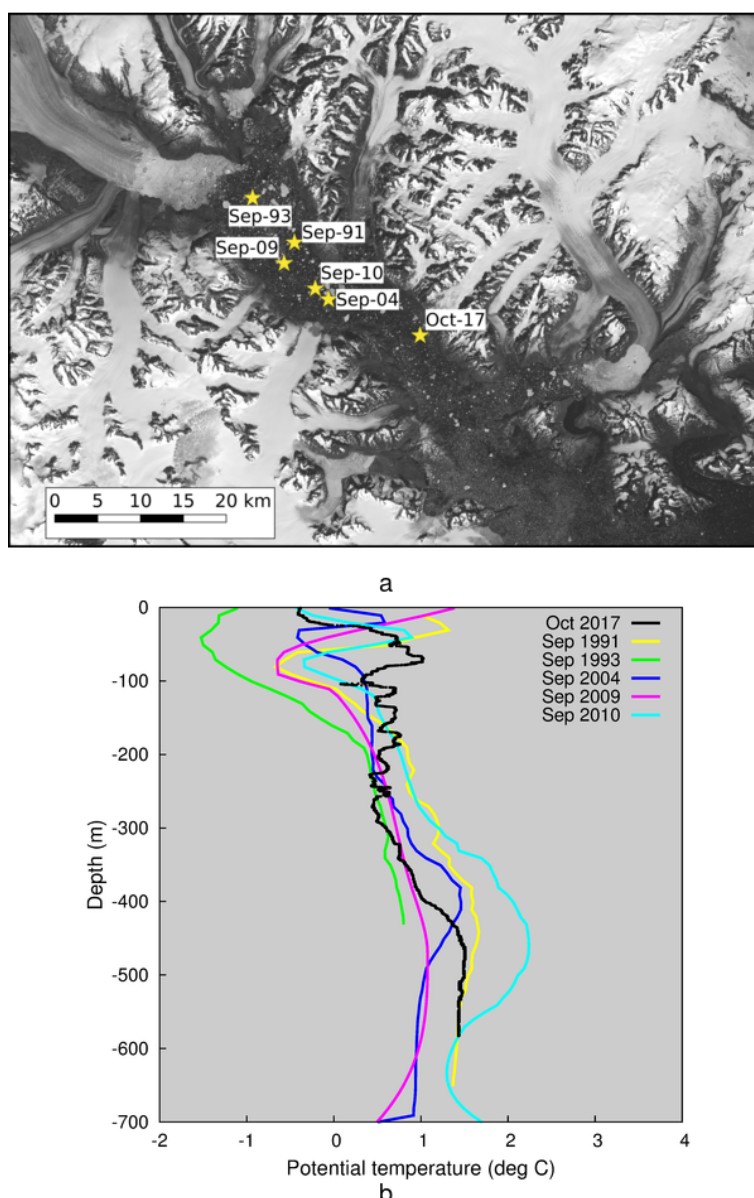

**Figure 5.** a) Location map (background Landsat Image 18/06/2016) and b) temperature profiles from published Conductivity Temperature Depth (CTD) data. September 1991 (Station K9, Andrews et al., 1994); September 1993 (Station KF3, Azetsu-Scott and Syvitski, 1999); September 2004 (Dowdeswell, 2004); September 2009 (Straneo et al., 2012); September 2010 (Inall et al., 2014); October 2017 (OMG Mission, 2016).

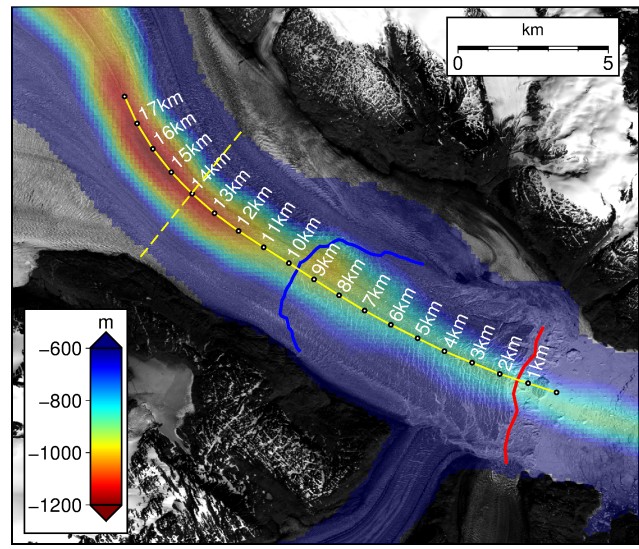

**Figure 6.** Glacier bed mapped using IceBridge BedMachine Greenland, Version 3 (Morlighem et al., 2017). The yellow profile and red and blue front positions relate to those drawn in Fig. 7. The dashed yellow transect indicates the line along which surface elevations were averaged for Fig. 3c.

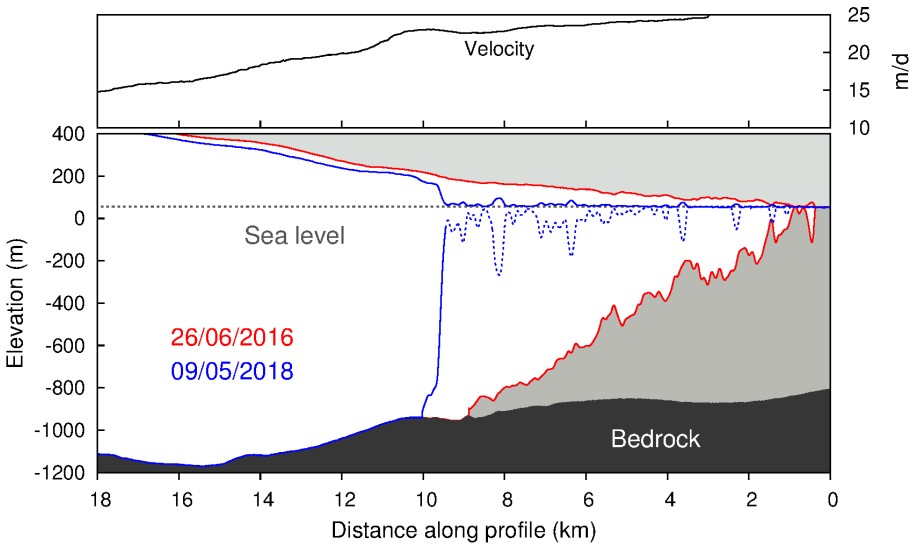

**Figure 7.** Surface elevations along the profile marked in Fig. 6 for advanced (red) and retreated (blue) front positions, corresponding to the red and blue points plotted in Fig. 3c. Bed (IceBridge BedMachine Greenland, Version 3 data, Morlighem et al., 2017) and surface velocities (based on TerraSAR-X data from 29/05/2014 to 09/06/2014) are along the same profile.

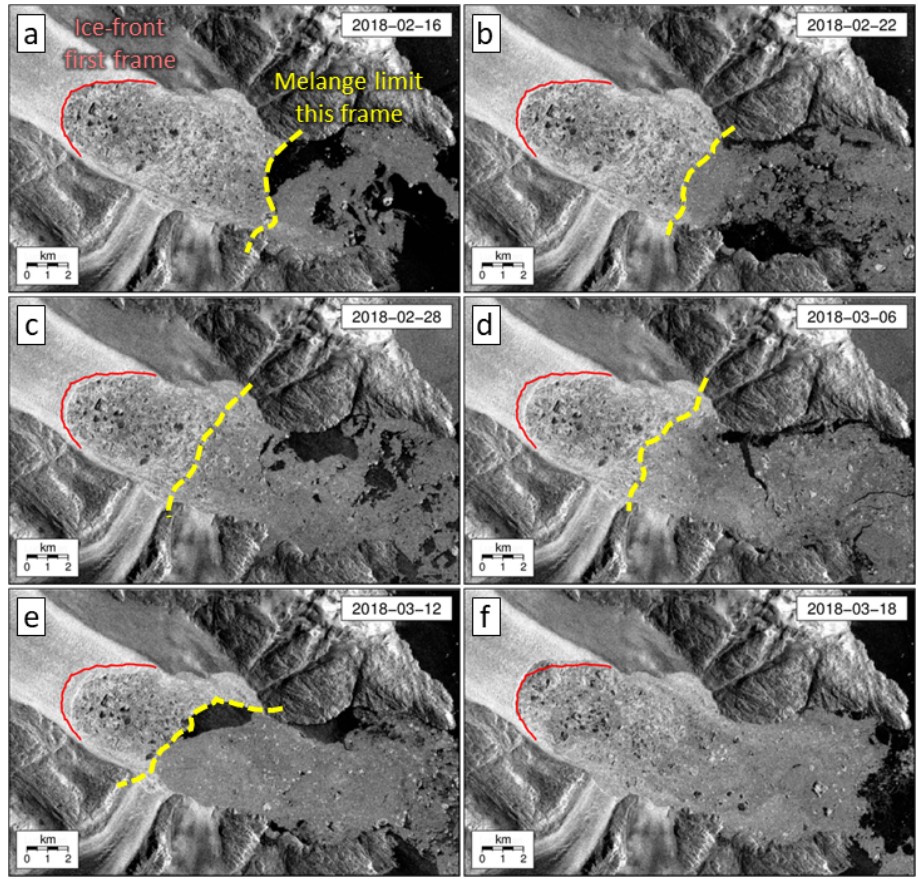

**Figure 8.** Sequence of Sentinel-1 SAR images of the Kangerdlugssuaq ice-front and fjord showing how the ice advances when mélange inhibits calving (panels a) to e)), then retreats again by calving because of loss or weakening of the mélange. The red line in each panel marks the ice-front position at the start of the sequence, and the yellow line marks the approximate extent of coherent mélange at each step. This behaviour is commonly seen in summer, but has recently also occurred in winter leading to ongoing glacier retreat.

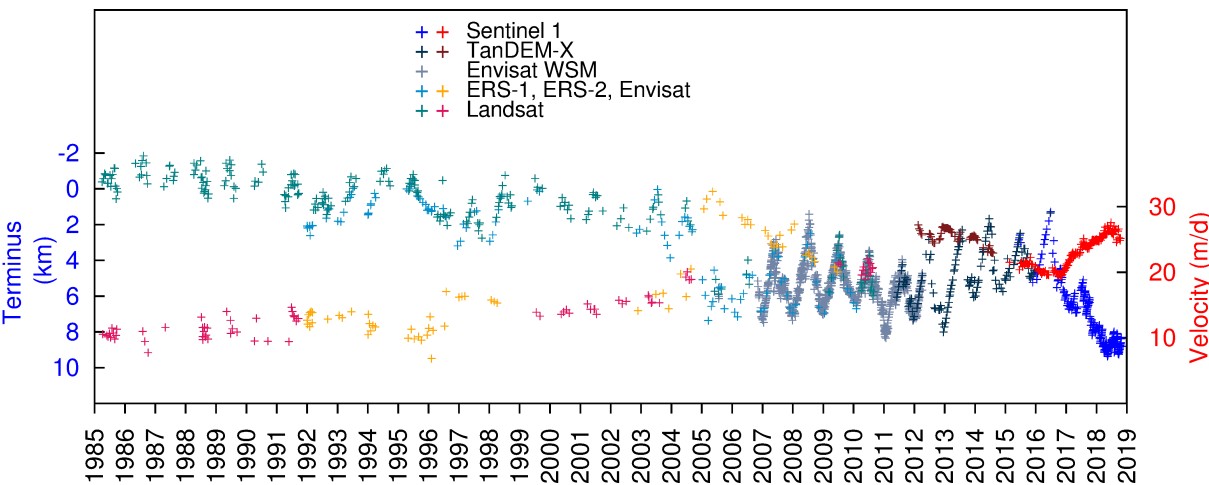

**Figure A1.** Time series of ice-front positions and velocity, with observations colour coded according to data source. Sentinel 1 includes 1A and 1B, TerraSAR-X refers to a single image from the TanDEM-X pair. Envisat WSM is Wideswath Mode data, European Remote Sensing satellites ERS-1 and ERS-2, and Envisat are Image Mode data, Landsat includes data from satellites 5, 7 and 8.

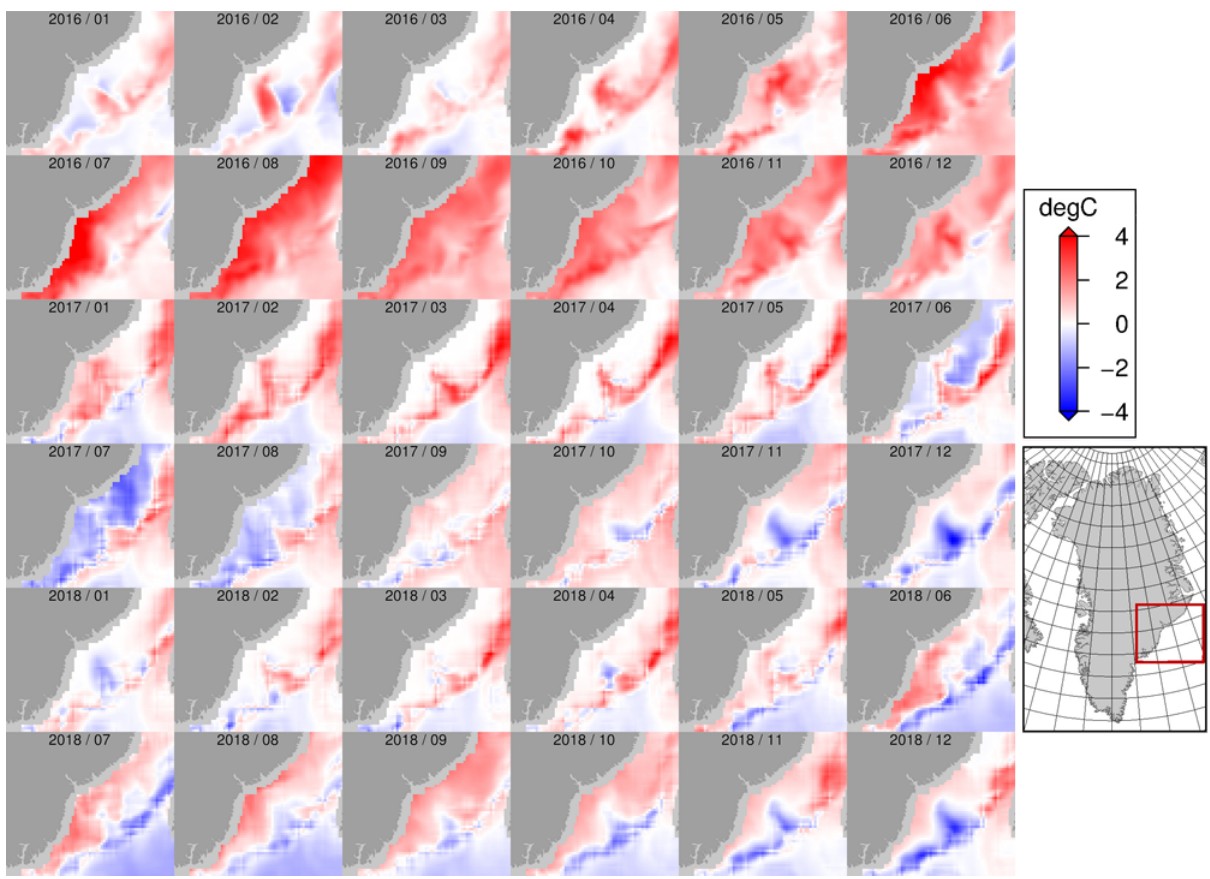

**Figure A2.** Anomalies in 5 m ocean potential temperatures. Temperatures for 1991–2017 are the Arctic Ocean Physics Reanalysis monthly mean data supplied by the Copernicus Marine Environment Monitoring Service (CMEMS). Temperatures for 2018 are based on monthly means of the Arctic Ocean Analysis and Forecast Product also from CMEMS. Anomalies are relative to the period 1992–2018.