# Peer review of "Warming of southeast Greenland shelf waters in 2016 primes large glacier for runaway retreat"

_The Cryosphere, 2018_

## Referee Comment (RC1) · Fraser (Referee) · 21 Jan 2019

Fraser (Referee)

neil.fraser@sams.ac.uk

The paper uses remote sensing data to generate an up-to-date time series of the Kangerdlugssuaq Glacier (KG) frontal position, thickness and flow velocity. The glacier has recently destabilised and is likely to retreat further in the near future due to sloping bedrock topography. Glacier destabilisation is attributed to anomalously warm surface waters breaking up ice melange which otherwise inhibits winter calving.

The paper is very well written with a clear, direct message and nice figures/animation. The work is highly topical and the data evidencing the latest KG destabilisation should be published promptly. The proximity of the grounding line to a region of retrograde bed slope is a particularly important point for predicting future mass loss. The authors

put forward a compelling mechanism to explain the recent retreat, ultimately attributing it to high sea surface temperatures on the shelf in 2016. However, I think more rigour is required to definitively establish a causal connection there, otherwise the proposed mechanism should be qualified somewhat.

I am not a glaciologist, but a physical oceanographer with expertise in this region. I can therefore only review the work from this viewpoint and can provide little feedback on, for example, the remote sensing or glacial dynamics aspects of the study.

Specific comments

1. The high sea surface temperatures in Kanger Trough in 2016, melange breakup episodes and destabilisation of KG could all be driven by a common forcing, namely either increased subsurface ocean heat content or increased atmospheric heating/solar radiation. Increased temperatures at depth will increase melting/calving at the terminus and, in particular, may have destabilised the terminus via undercutting. Increased atmospheric/solar heating would act to increase runoff and strengthen estuarine circulation, drawing shelf waters towards KG terminus at a greater rate. In both these scenarios, the breakup of melange may have played a role in KG's destabilisation or may simply be an additional symptom of the large scale forcing. You need to eliminate these possibilities before you can attribute the recent retreat and acceleration of KG to sea surface temperatures alone.

2. Can you be more quantitative about the forces involved with the melange stress bridges? You cite the Burton 2018 paper, but I think it would be appropriate to translate some of those results to this study as it is a crucial part of the proposed mechanism. Can you say that this is a more effective mechanism for mass loss than, for example, terminus undercutting (i.e. in ablation rate per Watt). Clearly the stress bridges are not strong enough to prevent terminus advance during a "normal" winter. It is not then clear that they may remain intact during these periods and prevent calving. I couldn't see any evidence of these stress bridges in Figure 4 or the animation, with icebergs seemingly

spaced apart. I acknowledge the image resolution may prevent smaller icebergs being distinguished from the surrounding sea ice. You say on P2L26 that the raw image data has an 8-30 m resolution. Could you use this to generate a zoomed-in, high-resolution image of the melange highlighting the potential for stress bridges?

3. I think the paper could say a little bit more about the mechanisms for exchange between fjord and shelf. Particularly, how do the warm surface waters on the shelf penetrate the fjord and flow all the way to the melange/terminus area. Simulations of Kanger fjord (e.g. by Cowton (2016) and Fraser (2018c)) indicate the net flow in the surface layer is out-fjord. This is not to say that surface layer inflow is impossible, and there is a reasonably large literature on fjord-shelf exchange in SE Greenland which could be referred to in order to fill-in this part of the picture.

4. I am not convinced the residence times within KF are sufficient for these warm surface waters to remain on annual to inter-annual timescales. It is a very dynamic environment with wintertime along-shelf winds driving rapid exchange events which punctuate the background circulation. Several estimates for the exchange through the fjord mouth exist (e.g. Sutherland 2014, Cowton 2016, Fraser 2018), and these can be used to generate back-of-the-envelope fjord flushing times on the scale of a few weeks or months. There are also regular katabatic winds driving strong outflow at the surface, described by Spall (2017) for Sermilik Fjord (I'm sure this can be extended to KF). Furthermore, the surface layer will be cooled by the atmosphere hence the formation of sea ice. I therefore cannot picture a scenario in which warm surface water simply "hangs around" in KF. It must be supplied via advection from the shelf or delivered from depth via either vertical mixing or buoyant overturning near the terminus/icebergs.

Technical corrections

P1L23: Should "fjord and shape" read "fjord size and shape"?

P2L8: I'd lose the word "ice" as it is implicit in the acronym "GrIS".

P4L21: "WGS84" had a space earlier on line 12.

P6L2: See specific comment 2.

P6L34: Were the subsurface or atmospheric temperatures anomalously warm during this period?

P7L5: See specific comment 4.

Figure 3: Most of this information is already in Figure 2. Perhaps they could be combined e.g. by making this an inset.

Figure 6 caption: "The dashed yellow line marks indicates the . . .". Lose either "marks" or "indicates".

Animation: I like this a lot, it communicates the severity of the recent retreat really well. But it would be nice to download it and be able to go through frame by frame.

Overall, a succinct and well written paper which I have enjoyed reading and reviewing. Another fascinating twist in the tale of Kangerdlugssuaq Glacier! Kind Regards, Neil Fraser

Please also note the supplement to this comment:
https://www.the-cryosphere-discuss.net/tc-2018-260/tc-2018-260-RC1-supplement.pdf

---

## Referee Comment (RC2) · Anonymous Referee #2 · 11 Mar 2019

Review of "Warming of SE Greenland shelf waters in 2016 primes large glacier for runaway retreat" by Bevan et al. (2019), The Cryosphere Discussions.

This paper uses a suite of remotely-sensed observations to show that Kangerdlug-gsuaq Glacier (KG) in southeast Greenland experienced substantial retreat during 2017–2018. This is important as KG may soon transition to a retrograde bed, which could lead to further inland migration. The authors provide observations that suggests a weakening of the winter ice mélange during this period, which they attribute to anomalous warming of near-surface shelf waters during 2016–early 2017. They then conclude that warm near-surface shelf waters weakened the ice mélange, altered the seasonal calving cycle, and triggered terminus retreat.

This paper is generally well written, and the time series of remotely-sensed observa-

tions will be highly useful to the community. However, in its present form, the oceanographic component of the paper is too speculative in its attribution to the mechanisms that inhibited/weakened the ice mélange and caused terminus retreat. In general, the authors should qualify their statements more (or provide quantitative evidence for their conclusions), along with considering all likely mechanisms for the observed retreat.

Major comments:

1. The authors propose that anomalously warm near-surface shelf waters during 2016–early 2017 reached the inner fjord, weakening the ice mélange. However, this result is only valid if the near-surface temperature variability on the shelf is directly transported to the inner fjord without significant damping. Do you have further evidence that these near-surface waters retained their anomalous heat content during their transit from the shelf to the inner KG fjord?

2. The authors should discuss how much up-fjord heat transport in the near-surface layer would be needed to substantially melt or inhibit the ice mélange. How does this heat transport compare to previous ocean modeling work (e.g., Cowton et al., 2016) and observations/theory (Sutherland et al., 2014; Jackson et al., 2016)?

3. Was there a coincident anomalous signal in subsurface ocean temperature, air temperature, or ice sheet runoff? These processes should also be considered/discussed as possible mechanisms for destabilizing the ice mélange/terminus.

Minor comments:

Page 1, L1: dash is not needed in "south-east" here or throughout the manuscript.

Page 1, L3: the statement "Here we show that the current retreat was driven" is too strong for the level of analysis presented in this manuscript. Please rephrase.

Page 1, L11: dash not needed in "run-off".

Page 1, L17: remove "specific".

Page 1, L23: change "glacier geometry, fjord and shape," to "glacier and fjord geometry".

Page 2, L5: add reference for Sutherland et al. (2014) (JGR: Oceans).

Page 2, L12: change "is currently" to "has currently".

Page 6, L6: It would be clearer to use "fjord mouth" instead of "down-fjord end".

Page 6, L9: change "thus reflects" to "could reflect".

Page 6, L20: change "in to" to "into".

Page 6, L23: change "meaning that it is well situated to interfere with" to "which could possibly inhibit".

Page 6, L27: change colon to semicolon.

Page 6, L31–32: this statement is too strong, please change the language to reflect your descriptive analysis.

Figure 1: dash is not needed in "re-analysis".

Figure 2, lower panel: do you have estimates of the spatial variability (i.e., show the standard deviation) in mean near-surface ocean temperatures from the reanalysis product?

Figure 5: it would be helpful to thicken the lines on the 2017 OMG CTD profiles

---

## Author Comment (AC1) · 17 Apr 2019

Dear reviewers, we attach a response to your comments and also a revised manuscript with highlighted changes. best wishes, Suzanne Bevan and co-authors.

Please also note the supplement to this comment:
https://www.the-cryosphere-discuss.net/tc-2018-260/tc-2018-260-AC1-supplement.zip

---

## Editor Comment (EC1) · Ginny Catania (Editor) · 19 Jul 2019

First, my apologies for this taking longer than it should have but I was travelling at a time that coincided with your re-submission. I want to provide a more thorough review of your manuscript, but I suspect that something went wrong when you updated Figure 3. The caption states: "The vertical bars mark dates when the feature-tracking metric indicated the presence (grey) or absence (pink) of a rigid mélange." However, I do not see that in the figure. I wonder if you could update the figure so that I can finish the review as I think my decision would hinge on these results. Thanks very much, Ginny

---

## Author Comment (AC2) · 19 Jul 2019

Dear Ginny,

thank you for finding time to reread the manuscript and please accept my apologies. You are correct, Fig. 3 was missing details of melange.

Please find attached a new fig. 3.

If you need me to incorporate it into a new manuscript pdf please let me know.

best wishes,

Suzanne Bevan

[Figure]

[Figure]

[Figure]

**Fig. 1.**

---

## Author Response (AR1)

Response to Reviewer 1 comments.

Dear Reviewer,

Thank you for your complimentary review and your thoughtful comments, we hope we have addressed them thoroughly. We have included a copy of the revised manuscript with changes highlighted.

Specific comments

*1. The high sea surface temperatures in Kanger Trough in 2016, melange breakup episodes and destabilisation of KG could all be driven by a common forcing, namely either increased subsurface ocean heat content or increased atmospheric heating/solar radiation. Increased temperatures at depth will increase melting/calving at the terminus and, in particular, may have destabilised the terminus via undercutting. Increased atmospheric/solar heating would act to increase runoff and strengthen estuarine circulation, drawing shelf waters towards KG terminus at a greater rate. In both these scenarios, the breakup of melange may have played a role in KG's destabilisation or may simply be an additional symptom of the large scale forcing. You need to eliminate these possibilities before you can attribute the recent retreat and acceleration of KG to sea surface temperatures alone.*

We fully acknowledge these points and have rewritten Section 3.2 to discuss the possible alternative drivers of retreat and included plots of surface air temperature. In summary, we say that the air temperature data show that the end of 2016, early 2017 and early 2018 were exceptionally warm and may have weakened the ice mélange. AW warming has been gradual, with no particular trigger in 2016, and these waters generally remain below 50 m so have minimal effect on winter sea ice formation.

We are convinced from the close timing of mélange breakup prior to calving, that terminus melt/undercut is not the main cause of recent instability of KG. The winter timing of the air temperature anomalies also suggests that the influence is more likely to be on sea ice, than on increased summer runoff driving enhanced buoyancy driven circulation driving frontal melt.

*2. Can you be more quantitative about the forces involved with the melange stress bridges? You cite the Burton 2018 paper, but I think it would be appropriate to translate some of those results to this study as it is a crucial part of the proposed mechanism. Can you say that this is a more effective mechanism for mass loss than, for example, terminus undercutting (i.e. in ablation rate per Watt). Clearly the stress bridges are not strong enough to prevent terminus advance during a "normal" winter. It is not then clear that they may remain intact during these periods and prevent calving. I couldn't see any evidence of these stress bridges in Figure 4 or the animation, with icebergs seemingly spaced apart. I acknowledge the image resolution may prevent smaller icebergs being distinguished from the surrounding sea ice. You say on P2L26 that the raw image data has an 8-30 m resolution. Could you use this to generate a zoomed-in, high-resolution image of the melange highlighting the potential for stress bridges?*

We have expanded on the discussion of the ice mélange role in preventing terminus advance and quoted the theoretical estimates of the forces involved. At your suggestion we have also included and described a higher resolution image of the front. We do not calculate actual ablation rates but cite two papers which demonstrate that the potential for ice mélange to impose a seasonal signal on frontal advance is much greater than that for submarine melt. Please see Section 3.1.

*3. I think the paper could say a little bit more about the mechanisms for exchange between fjord and shelf. Particularly, how do the warm surface waters on the shelf penetrate the fjord and flow all the way to the melange/terminus area. Simulations of Kanger fjord (e.g. by Cowton (2016) and Fraser (2018c)) indicate the net flow in the surface layer is out-fjord. This is not to say that surface layer inflow is impossible, and there is a reasonably large literature on fjord-shelf exchange in SE Greenland which could be referred to in order to fill-in this part of the picture.*

The rewritten Section 3.2 now contains more about fjord—shelf exchange mechanisms. Please see the second paragraph of Section 3.2.2.

*4. I am not convinced the residence times within KF are sufficient for these warm surface waters to remain on annual to inter-annual timescales. It is a very dynamic environment with wintertime along-shelf winds driving rapid exchange events which punctuate the background circulation. Several estimates for the exchange through the fjord mouth exist (e.g. Sutherland 2014, Cowton 2016, Fraser 2018), and these can be used to generate back-of-the-envelope fjord flushing times on the scale of a few weeks or months. There are also regular katabatic winds driving strong outflow at the surface, described by Spall (2017) for Sermilik Fjord (I'm sure this can be extended to KF). Furthermore, the surface layer will be cooled by the atmosphere hence the formation of sea ice. I therefore cannot picture a scenario in which warm surface water simply "hangs around" in KF. It must be supplied via advection from the shelf or delivered from depth via either vertical mixing or buoyant overturning near the terminus/icebergs.*

This is a fair comment and we have now changed this paragraph, removing the suggestion that the warm water would persist into the winter. We add that mild air temperatures may have inhibited sea ice formation. See the last paragraph of Section 3.2. Also note that on learning that it is probably unreliable, we have removed the inner fjord CTD (Oct-17-B) profile from the plots

*Technical corrections*
*P1L23: Should "fjord and shape" read "fjord size and shape"?*

We have changed to 'glacier and fjord geometry'.

*P2L8: I'd lose the word "ice" as it is implicit in the acronym "GrIS".*

True, thanks.

*P4L21: "WGS84" had a space earlier on line 12.*

We now consistently use WGS84.

*P6L2: See specific comment 2.*

Addressed above.

*P6L34: Were the subsurface or atmospheric temperatures anomalously warm during this period?*

The 30 m, 50 m and 100 m temperatures follow a similar pattern to the 5 m temperatures. At 200 m the temperatures show a gradual warming over recent years but with no particular trigger in 2016. Please see the answer to point 1 and Sections 3.2.1 and 3.2.2.

*P7L5: See specific comment 4.*

Addressed above.

*Figure 3: Most of this information is already in Figure 2. Perhaps they could be combined e.g. by making this an inset.*

We would prefer not to do this as we think the overview in Fig. 2 is important and the time zoom in Fig. 3 makes it easier to relate variations in parameters such as surface elevation and frontal position relative to each other.

*Figure 6 caption: "The dashed yellow line marks indicates the : : :". Lose either "marks" or "indicates".*

We have removed 'marks' thanks.

Response to Reviewer 2 comments.

Thanks also for your comments and helpful advice and we have amended the text to be more qualified in some of our statements.

For example, in the abstract we have added the word 'probably' to "Here we show that the current retreat was probably driven by…". Also, in the last paragraph of the Introduction we have changed "We demonstrate that the mélange weakening…" to "We propose that the mélange weakening…".

Please see our other responses below.

*Review of "Warming of SE Greenland shelf waters in 2016 primes large glacier for runaway retreat" by Bevan et al. (2019), The Cryosphere Discussions.*
*This paper uses a suite of remotely-sensed observations to show that Kangerdluggsuaq Glacier (KG) in southeast Greenland experienced substantial retreat during 2017–2018. This is important as KG may soon transition to a retrograde bed, which could lead to further inland migration. The authors provide observations that suggests a weakening of the winter ice mélange during this period, which they attribute to anomalous warming of near-surface shelf waters during 2016–early 2017. They then conclude that warm near-surface shelf waters weakened the ice mélange, altered the seasonal calving cycle, and triggered terminus retreat.*

*This paper is generally well written, and the time series of remotely-sensed observations will be highly useful to the community. However, in its present form, the oceanographic component of the paper is too speculative in its attribution to the mechanisms that inhibited/weakened the ice mélange and caused terminus retreat. In general, the authors should qualify their statements more (or provide quantitative evidence for their conclusions), along with considering all likely mechanisms for the observed retreat.*

*Major comments:*

*1. The authors propose that anomalously warm near-surface shelf waters during 2016– early 2017 reached the inner fjord, weakening the ice mélange. However, this result is only valid if the near-surface temperature variability on the shelf is directly transported to the inner fjord without significant damping. Do you have further evidence that these near-surface waters retained their anomalous heat content during their transit from the shelf to the inner KG fjord?*

We do not have any further evidence that near-surface waters retained their heat content but in response to Reviewer 1, comment 3 we have now included more text discussing fjord—shelf exchange mechanisms. Please see the text in Section 3.2.2.

*2. The authors should discuss how much up-fjord heat transport in the near-surface layer would be needed to substantially melt or inhibit the ice mélange. How does this heat transport compare to previous ocean modeling work (e.g., Cowton et al., 2016) and observations/theory (Sutherland et al., 2014; Jackson et al., 2016)?*

Unfortunately we do not have the resources to answer this question but emphasise the correspondence in timing between warm surface waters in the fjord and a weakening and dispersal of wintertime ice mélange. Quantitative heat budget calculations would be a good topic for further work.

*3. Was there a coincident anomalous signal in subsurface ocean temperature, air temperature, or ice sheet runoff? These processes should also be considered/discussed as possible mechanisms for destabilizing the ice mélange/terminus.*

In answer to the comments of Reviewer 1, we now also discuss the possibility that deeper subsurface ocean temperatures or air temperatures may have contributed to destabilizing the mélange and terminus. We find that air temperatures were anomalously warm (see Figs. 2 and 3) at the start of 2017 and 2018, this timing suggesting an impact on sea ice as opposed to runoff. Deeper ocean temperatures (200 m) have shown a gradual warming since 2010 but show no obvious trigger point in 2016/17. Please see the rewritten Section 3.2.

*Minor comments:*

*Page 1, L1: dash is not needed in "south-east" here or throughout the manuscript.*

This is a matter of style, journal or British versus US English, however, we have accepted your suggestion.

*Page 1, L3: the statement "Here we show that the current retreat was driven" is too*

*strong for the level of analysis presented in this manuscript. Please rephrase.*

Accepted, we have inserted the word probably.

*Page 1, L11: dash not needed in "run-off".*

Changed.

*Page 1, L17: remove "specific".*

OK, removed.

*Page 1, L23: change "glacier geometry, fjord and shape," to "glacier and fjord geometry".*

Changed.

*Page 2, L5: add reference for Sutherland et al. (2014) (JGR: Oceans).*

Added, thanks.

*Page 2, L12: change "is currently" to "has currently".*

We have changed the sentence to read 'Here we show that by the end of 2018 the calving front of Kangerdlugssuaq Glacier was further upstream…'

*Page 6, L6: It would be clearer to use "fjord mouth" instead of "down-fjord end".*

We do not really mean as far downstream as at the fjord mouth. We have changed to 'down-fjord edge'.

*Page 6, L9: change "thus reflects" to "could reflect".*

We changed to 'could thus reflect'.

*Page 6, L20: change "in to" to "into".*

Done.

*Page 6, L23: change "meaning that it is well situated to interfere with" to "which could possibly inhibit".*

We changed to 'means that it could inhibit'.

*Page 6, L27: change colon to semicolon.*

Changed.

*Page 6, L31–32: this statement is too strong, please change the language to reflect your descriptive analysis.*

We changed 'we are able to demonstrate' to 'we are able to provide further evidence'.

*Figure 1: dash is not needed in "reanalysis".*

We have removed the hyphen from all occurrences of reanalysis.

*Figure 2, lower panel: do you have estimates of the spatial variability (i.e., show the standard deviation) in mean near-surface ocean temperatures from the reanalysis product?*

We have calculated this and now plot the actual values rather than the anomalies (which are shown in Fig 3) with +/- one standard deviation indicated by shading. Also, now that the reanalysis products for 2017 are available so we have used these instead of analysis data.

*Figure 5: it would be helpful to thicken the lines on the 2017 OMG CTD profiles*

Done. Also note that we have removed the Oct-17-B profile as we are led to believe that it is probably unreliable.

[revised manuscript text omitted]

---

## Author Response (AR2)

Dear Editor,

thank you for your all of your comments and suggestions, in particular the request to improve the characterization of the mélange using a metric based on the coherence of feature tracking. This analysis has been very successful, supports our argument, and has greatly improved the manuscript.

We discuss below the points raised in this letter and on the marked-up pdf.

Given that this paper has been reviewed thoroughly both via Discussions and by yourself, we wonder whether it would be possible to avoid another round of reviews.

Regards,

Suzanne Bevan and co-authors.

Comments to the Author:

I have read your manuscript several times and I thank you for carefully addressing the reviewer comments. Unfortunately, I do not think your manuscript is ready for publication with The Cryosphere at this time for the following reasons:

1. While I remain open to the story presented in the manuscript - that warm surface waters weakened melange, I do not see how this idea is supported in the manuscript as it is presented. You simply have no quantified metric of melange rigidity to really prove your observations. A few times in the text you say that you "believe that warm surface waters weakened melange", but science must be more than belief - you must prove it. Other authors have used coherence of InSAR velocities in the melange to quantify rigidity of melange cover (Kehrl et al., 2018) or sea-ice fraction (Fried et al., 2018), or surface temperature (Cassotto et al. 2015) and so I recommend that you use one of these to really prove your conclusion.

*We have now included an assessment of mélange rigidity based on velocity mapping similar to Kehrl et al. (2017). This has worked really well and we are glad that you suggested it.*

*With regard to using SSTs or ice fraction products we have investigated the OSTIA sea ice fraction and SST products but they are of rather coarse (5 km) resolution to use in the actual fjord although they are adequate on the shelf. We therefore do not think they are a reliable evidence for conditions in the fjord itself.*

2. It is unclear why you focus on the most recent retreat as your conclusions could be strengthened if you also examined prior retreats (like 2004). If 2004 acted similarly then it would support your conclusion, but even if 2004 did not act similarly, it would allow you to put your results into some context. Perhaps you could also do this for the period in 1997 when no winter advance happened.

*Regarding the 2004 (actually early 2005) retreat we have now taken care to both highlight the temperature anomalies during this period and to refer to Christoffersen et. al. (2011,2012) work on this period. Please see P. 6, lines 25-28. Here we also refer to the lack of advance in 1996.*

3. Finally, I would like for you to revise the text to create separate Results and Discussion sections. I found it difficult to weed through to find you actual results.

*We have done as you suggest and hope this improves understanding.*

4. Other comments are in the attached pdf.

*We have considered all the comments on the pdf. Some suggestions we have adopted, and others have been addressed by including the feature-tracking based metric for mélange rigidity. Below we discuss the remaining comments.*

P3, L15. I'm not convinced that this is a significant observation.  One CTD cannot be used to draw much of a conclusion about the water in the fjord as a whole. Additionally, the presence of AW may be driven by runoff and in the fall, runoff has dwindled and thus AW may be less present.

*We propose to keep this reference to the CTD profiles as CTD observations are commonly used to characterize water in the fjord in spite of their limited sampling. The second point is a good point and it supports our conclusion that it was unlikely that increased AW presence and temperature was a key factor in limiting winter advance. We have added text to make this point so thank you for highlighting it.*

P4, L26. Do you think it's rigid only in this one region to the side of the main glacier terminus?  Is this sufficient to impact the glacier given that the compressive region is so small?

*With the added evidence of the feature tracking we have decided to remove this figure and reference to it.*

P7. It seems out of place to mention the elevation data so far into the text, when the figure is presented much earlier on.  I might suggest re-arranging the presentation of data to keep elevation and velocity data along with Fig. 7.

*As we have now separated the Results from the Discussions we are able to describe and refer to the elevation and velocity data sooner in the text.*

I hope you do not take this decision to heart. I think with a little work you can really prove your conclusions and revise this into a widely-cited paper. I encourage you to do so.

*We hope you agree that the extra analysis has improved the paper and thanks again for your encouragement.*

---

## Author Response (AR3)

Dear Editor,

Thank you very much for your thorough reading of the manuscript and for the minor revision suggestions. We have addressed them as follows.

We have changed the title to:

'Impact of warming shelf waters on ice mélange and terminus retreat at a large SE Greenland glacier'

Page 3, line 29. We have added the following sentence:

'The 5 m data were chosen to sample Polar Surface Water (PSW) and the 200 m data to sample the upper layers of Atlantic Water (AW) (Sutherland et al., 2014).

Page 5, line 15. The sentence now reads:

'In winters 2017 and 2018, when the glacier continued to calve, a rigid mélange was not maintained for more than one month in either year.

And we have moved the final sentence of this paragraph to the methods section. The white parts of the image in Fig. 3 are when no velocity maps were created. We have not been explicit about this as we think it is sufficient that the caption states 'The vertical bars mark dates when the feature-tracking metric indicated the presence (grey) or absence (pink) of a rigid mélange.'

Page 5, line 19.

You suggest that it would be helpful to plot or state the mean air temperatures. The plot shows anomalies from the monthly means so it does not make sense to plot the actual means on the same graph. Also, there is a different long-term mean for each month so we would need to list 12 different values. We hope you are satisfied with our reasons for making no changes in this case.

Page 5, last sentence. We have added:

'The profiles show relatively warm water at depth in September 2010 which is reflected in the re-analysis data at 200 m (Fig. 2c). However, whilst the October 2017 profile shows warm temperatures in the upper 100 m, the waters at depth are not as notably warm as the re-analysis data show out on the shelf.'

Page 6, line 13. We have added the references to Cassotto et al. (2015) and Fried et al. (2018).

Page 7, line 29. We have changed the phrase 'facilitated by the relatively deep sills' to 'unhindered by the relatively deep sills'.

We have added horizontal grid lines to Figs. 2b and 2c, and replaced Fig. 3 with the correct version.

The caption to Fig. 7 now reads:

'Surface elevations along the profile marked in Fig. 6 for advanced (red) and retreated (blue) front positions, corresponding to the red and blue points plotted in Fig. 3c. Bed (IceBridge BedMachine Greenland, Version 3 data, Morlighem et. al., 2017) and surface velocities (based on TerraSAR-X data from 29/05/2014 to 09/06/2014) are along the same profile.

Once again, many thanks for the effort you have put into helping us improve this manuscript.

Regards,

Suzanne Bevan and co-authors.

[revised manuscript text omitted]